## Research Article

competence; healthcare workers; assessment tools; developing countries; education

**Corresponding author:**
Gloria A. Pedersen;
Email: gapedersen@gwu.edu

# A mixed methods evaluation of a World Health Organization competency-based training package for foundational helping skills among pre-service and in-service health workers in Nepal, Peru and Uganda

Gloria A. Pedersen[1] , Pragya Shrestha[2], Josephine Akellot[3], Alejandra Sepulveda[4], Nagendra P. Luitel[2], Rosco Kasujja[5], Carmen Contreras[4,6,7], Jerome T. Galea[7,8] , Leydi Moran[4], Vibha Neupane[2], Damodar Rimal[2], Alison Schafer[9] and Brandon A. Kohrt[1]

[1]Center for Global Mental Health Equity, Department of Psychiatry and Behavioral Health, George Washington University, Washington, DC, USA; [2]Program Department, Transcultural Psychosocial Organization (TPO) Nepal, Kathmandu, Nepal; [3]Uganda Country Office, Programs Department, HealthRight International, Kampala, Uganda; [4]Socios En Salud Sucursal Perú, Lima, Peru; [5]School of Psychology, Department of Mental Health, Makerere University, Kampala, Uganda; [6]Harvard Global Health Institute, Harvard University, Cambridge, MA, USA; [7]School of Social Work, University of South Florida, Tampa, FL, USA; [8]Department of Global Health and Social Medicine, Harvard Medical School, Boston, MA, USA and [9]Department of Mental Health and Substance Use, World Health Organization, Geneva, Switzerland

## Abstract

Health systems globally demand more competent workers but lack competency-based training programs to reach their goals. This study evaluates the effectiveness of a competency-based curriculum (EQUIP-FHS) for trainers and supervisors to teach foundational helping knowledge, attitudes and skills, guided by the WHO/UNICEF EQUIP platform, to improve the competency of in-service and pre-service workers from various health and other service sectors. A mixed-methods, uncontrolled before-and-after trial was conducted in Nepal, Peru, and Uganda from 2020 to 2021. Trainees' ($N = 150$) competency data were collected during 13 FHS trainings. Paired *t*-tests assessed pre- to post-change in ENACT competency measures (e.g., harmful, helpful). Qualitative data was analyzed using thematic analysis. EQUIP-FHS trainings, on average, were 20 h in duration. Harmful behaviors significantly decreased, and helpful behaviors significantly increased, across and within sites from pre- to post-training. Qualitatively, trainees and trainers promoted the training and highlighted difficult competencies and areas for scaling the training. A brief competency-based curriculum on foundational helping delivered through pre-service or in-service training can reduce the risk that healthcare workers and other service providers display harmful behaviors. We recommend governmental and nongovernmental organizations implement competency-based approaches to enhance the quality of their existing workforce programming and be one step closer to achieving the goal of quality healthcare around the globe.

## Impact statement

Through the World Health Organization (WHO) and United Nations International Emergency Fund (UNICEF) Ensuring Quality in Psychological Support (EQUIP) initiative, a curriculum for trainers and supervisors to teach knowledge, attitudes and skills and assess foundational helping competencies was iteratively developed in 2020. Each module includes an evidence-based competency assessment, therein supporting a trainer or supervisor in targeting, evaluating, and providing feedback on competencies throughout the training and proactively adapting the plan as needed to meet trainees' goals. This study looks to determine if the WHO's curriculum can improve foundational helping competencies of different in-service (obstetricians, nurses, community health workers) and pre-service (public health, social work, nursing, and psychology students) workers from Nepal, Peru and Uganda. Before training, most healthcare workers and other service providers were scored as "potentially harmful" on foundational helping competencies. After completing the training, they displayed significantly fewer harmful behaviors and significantly more helpful behaviors. Our study found that trainees and trainers valued the

EQUIP-FHS curriculum in its content and approach, highlighting the use of competency-based role plays and feedback in helping trainees achieve their competency goals. Based on the qualitative and quantitative results of this study, we strongly suggest that government, non-profit, and academic organizations implement competency-based training and assessment.

## Introduction

Foundational helping competencies are the core behaviors that help to strengthen relationships, build emotional well-being and support positive interactions for people providing services across the health and community service fields (Watts et al., 2021). Foundational helping competencies involve the necessary knowledge, attitudes and skills for helping, and are based upon common factors in mental health and psychological services, which have been widely researched and identified as essential and universal pre-requisite for the effective delivery of any psychosocial or psychological components in health and global mental health interventions and align with the global competency framework for Universal Health Coverage (UHC) (Imel and Wampold, 2008; Wampold, 2011, 2015; Singla et al., 2017; Pedersen et al., 2020; World Health Organization, 2022). Other foundational helping competencies include rapport building, the demonstration of empathy, using culturally or age-appropriate terminology and concepts for distress, and ensuring communication of hope. Competent use in foundational helping by health and care workers improves outcomes for people accessing different fields of health services, ranging from surgery to pain clinics (Kohrt et al., 2015; Hojat, 2016; Golshan et al., 2019; Walsh et al., 2019; Surchat et al., 2022).

For instance, Lambert (2022) discusses the significance of communication for surgeons to support the palliative management of patients and considers the promotion of hope to be "the most necessary element of care for most patients and surgeons in the palliative setting." Similarly, a review by Heyn et al. (2023) explored how expressing positive emotions during interactions could support the relationship of the practitioner–patient (nurse–patient, physician–patient, professional–patient). They identified several strategies and mechanisms, including using open/honest communication, genuineness, empathy, contributing to a patient's hope for healing, forming trust, offering reassurance and comfort, providing emotional support through positive affirmation or "praise and support" and creating a feeling of connectedness with the patient. Research has also shown a lack of foundational helping competencies could impact patient outcomes or affect treatment adherence. For instance, Tiwary et al. (2019) provide two case reports where healthcare professionals' poor communication, such as failing to check the patient understands the diagnosis and treatment plan or appropriately involving and communicating to the patient's family to support treatment adherence, could be what led to life-threatening complications with patients

in Nepal. Additionally, Blasi et al.'s (2001) systematic review found that healthcare consultations that provided cognitive care (influencing patients' beliefs about the effects of treatment) and emotional care (being warm, empathic, or reassuring) had significant impacts on patient outcomes in decreasing pain and increasing the speed of recovery compared to neutral consultations.

Demands are increasing to improve the quality of the health, care, social, and other service workforce to reach UHC, and competency-based strategies for training and supervision have been proposed as a pathway to get there (Frenk et al., 2010; Cometto et al., 2020; Uribe, 2022). More competency-based training curricula that address foundational helping competencies are needed. Through the World Health Organization (WHO) and United Nations International Emergency Fund (UNICEF) Ensuring Quality in Psychological Support (EQUIP) initiative (Kohrt et al., 2020) (www.equipcompetency.org), a curriculum for trainers and supervisors to teach knowledge, attitudes and skills and assess foundational helping competencies, called *Teaching Foundational Helping Skills: An EQUIP Competency-Based Training Manual for Trainers and Supervisors* (EQUIP-FHS) was iteratively developed in 2020. To minimize training burden and maximize effectiveness, the curriculum is designed as a modular competency-based training rather than a "one-size-fits-all" approach and intends to be feasible for implementation in low-resource settings. Each module is complemented by an evidence-based competency assessment using items from the ENhancing Assessment of Common Therapeutic (ENACT) factors tool (Kohrt et al., 2015), found on the EQUIP platform, therein supporting a trainer or supervisor in targeting, evaluating, and providing feedback on foundational helping competencies throughout the training (see Figure 1).

This EQUIP competency-based approach to teaching foundational helping and assessing competencies allows trainers and supervisors to proactively adapt the training plan and provide tailored feedback based on trainees' existing and developing competencies. Like the tools and resources on the EQUIP platform, the EQUIP-FHS curriculum is intended to be freely available for trainers and supervisors and it can potentially improve foundational helping competencies among all health professionals and other service providers.

The present research aims to evaluate the effectiveness of the EQUIP-FHS curriculum to improve competency from pre- to post-training of different pre-service and in-service healthcare workers

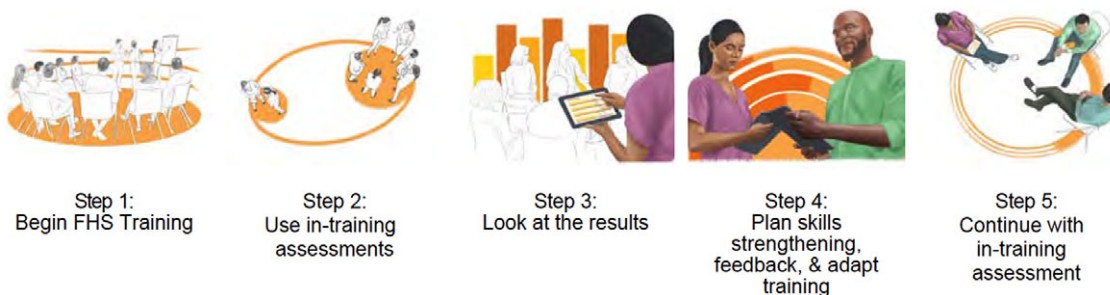

Step 1:
Begin FHS Training

Step 2:
Use in-training assessments

Step 3:
Look at the results

Step 4:
Plan skills strengthening, feedback, & adapt training

Step 5:
Continue with in-training assessment

**Figure 1.** Steps for an EQUIP-FHS competency-based training with the ENhancing Assessment of Common Therapeutic factors (ENACT) tool.

and other service providers who have no prior experience delivering formal mental health or psychosocial interventions (e.g., obstetricians) and which are working in low-resourced settings. Additionally, the current study will assess trainees' and trainers' acceptability and perceived benefit of partaking in an EQUIP-FHS training.

## Methods

### Context

This study was conducted in Nepal, Peru, and Uganda from 2020 to 2021. Trainings were conducted remotely, in-person, or using a hybrid method according to existing COVID-19 policies within each setting. In Kampala, Uganda, EQUIP-FHS trainings were implemented in-person by a team with HealthRight Uganda and Makerere University. In Kathmandu, Nepal, trainings were implemented by the Transcultural Psychosocial Organization (TPO) Nepal using a hybrid method, wherein TPO Nepal provided conference room spaces with WIFI and Zoom connections for small groups of trainees to join remotely while staying socially distanced. In the Metropolitan Area of Lima, Peru, the implementing organization Socios en Salud (SES) delivered all trainings remotely over Zoom. Trainees in Peru without access to unlimited WIFI were lent cell phones with unlimited internet.

### Design

The current study is based on the EQUIP-Foundational Helping Skills (EQUIP-FHS) study, a multi-site, mixed methods, uncontrolled before-and-after trial (*ClinicalTrials.gov ID: NCT04511156*). The study used an intervention mixed methods framework, wherein qualitative data were collected primarily to explain results of the training outcomes and to understand contextual factors during the intervention that could affect the outcome, including trainers' and trainees' perceptions of the feasibility, acceptability and perceived utility of the training (Fetters et al., 2013). The objective was to demonstrate that a brief, competency-based training intervention in foundational helping knowledge, attitudes and skills will improve the competency of workers naive to mental health and psychosocial services. We integrate quantitative and qualitative data at the interpretation and reporting level through narrative using a contiguous approach wherein quantitative results are reported first, followed by qualitative results. Integration is further elaborated via in-text reference and in the discussion.

### Intervention description

The EQUIP-FHS curriculum was developed in a modular format, with each module relating to a specific foundational helping competency (e.g., nonverbal communication, confidentiality). The content for the pilot-test version of the training manual used in this study was developed collaboratively and iteratively, led by EQUIP team members at the George Washington University (GWU) and the WHO with support from a review group of selected field and academic experts between August to November 2020. Description of this first phase of module development can be found in Supplementary Box S1.

The training outline implemented by the sites is in the Supplementary Material. It includes approximately 2 days of EQUIP-FHS modules and approximately a half-day of training that involves remediation, "Skills Strengthening," of any foundational helping competencies that have been identified via the assessments or which trainers felt needed extra time for review. Topics covered in the EQUIP-FHS modules align with

competencies found in ENACT. During a pilot test, trainers deemed certain competencies as more advanced in mental health and psychosocial services and, therefore, less relevant. The ENACT items related to these modules included social functioning (Item 08), explanatory models (Item 09), family involvement (item 10), coping mechanisms (Item 13), psychoeducation (Item 14) and eliciting feedback (Item 15). As such, it was agreed among the trainers across the sites that they could integrate these topics into other modules when it was seen as complementary, such as including didactic on psychoeducation and explanatory models when teaching the module on promoting hope (ENACT Item 12). The decision to integrate typically relied on how trainees were progressing in the initial modules, using trainers' subjective judgment and objective competency assessment scores.

All trainers accessed assessment results daily to adapt training plans as they saw fit and to give competency-based feedback to trainees. Trainers had access to an EQUIP e-learning module, "Giving and Receiving Feedback" (https://equipcompetency.org), as an optional guide to support competency-based feedback. Finally, as a supplementary training tool, each site had access to brief ("mini") roleplay videos (1–2 min each) that show helpful and unhelpful examples of a given ENACT competency. Videos were initially developed in English, then recreated in Nepali and Spanish by the partnering sites.

### Cultural adaptation and translation

The EQUIP-FHS curriculum and assessment tools were translated into the respective languages that the training would be conducted: Nepali in Nepal, Spanish in Peru, and English in Uganda. Before running any trainings, each site ran a small pilot training with a group of mental health and psychosocial service experts ($N = 3$–6) to support contextual adaptation and ensure relevancy for healthcare workers and other service providers that do not have experience with mental health services. A charting form (by EQUIP-FHS module, per activity) was used to track adaptations made by each site to support cross-site learning and consistency. Adaptations were primarily made to meet trainees' needs to support learning and facilitation and in line with using a modular approach to training, rather than cultural-specific adaptations.

For instance, in Uganda, there was only one trainer, so trainees were asked to volunteer in role plays with the trainer for demonstrative purposes during training activities rather than a second trainer. In Peru, they took the modular approach of introducing Module 5 "Attitudes toward helping" at the beginning of the training and offered more examples of the importance of mental health in health care to promote engagement and confidence among the in-service trainees like the CHWs and obstetricians. In Nepal, the existing acronyms were useful tools for synthesizing learning, so the trainers added more during certain modules, such as adding EAR: Empathy, Acceptance and Reflection into the verbal communication module. In Uganda and Nepal, trainers allocated more time in the verbal communication module for roleplay practice and review on changing close-ended to open-ended questions.

### Instrument

*ENhancing Assessment of Common Therapeutic factors (ENACT)* rating tool is a competency assessment tool based on common factors in mental health and psychological support. Each item relates to a foundational helping competency. It is used in objective structured clinical examinations, that is, standardized role plays

with a simulated client (actor). It is a 15-item tool with an ordinal scale; each competency item has four scoring levels: Level 1 = potentially harmful; Level 2 = not done or limited demonstration of basic competency; Level 3 = basic helping competency; Level 4 = advanced helping competency. For each competency, approximately 8–12 individual attributes can be dichotomously checked (0,1) to determine the level.

### Participant recruitment

A total of 160 trainees participated in 13 EQUIP-FHS trainings across the sites (three trainings each in Nepal and Uganda, seven trainings in Peru). All trainees participating were over 18 years old, had no experience in mental health or psychosocial service delivery, and had fluency in the language in which the trainings would be conducted: Nepali in Nepal, Spanish in Peru, and English in Uganda. Trainees who were under 18 years old and had experience with mental health service delivery or similar (e.g., completed psychology degree) were excluded. In Nepal, trainees were recruited from local universities. In Uganda, trainees were recruited from the HealthRight Organization. In Peru, trainees were recruited through a partnership with the Bonilla-La Punta Health Network, health establishments of the Integrated Health Network of Lima Norte (DIRIS Lima Norte) and with the Private Universities of Lima.

### Research and training design

Pre- and post-training research days were used to collect primary and secondary outcomes with support from research staff (trained raters, actors, and qualitative researchers). A structured role play ENACT assessment was measured pre-training (day before) and post-training (either on the last day of training or the following day). Each pre-post assessment was 10 min (per participant, per timepoint). Roleplay assessments were done remotely (e.g., over Zoom) for remote trainings and in-person for face-to-face trainings. All assessments were video recorded. Supplement Figure S2 shows the sequence of research and training activities.

### Quantitative analysis

A summary rating was calculated per assessment at the ENACT attribute and level measures. An algorithm in Excel supported this calculation. Description of the inter-rater reliability (IRR) processes and calculation of IRR and summary scores can be found in the Supplementary Material. Statistical analysis was conducted using R Statistical Software (2022) and RStudio Integrated Development Environment (v2022.12.0.353) (Posit team, 2022; R Core Team, 2022) with the tidyverse, dplyr, lessR and irr packages (Gamer et al., 2019; Wickham et al., 2019; Gerbing, 2021; Wickham et al., 2023). From an eligible sample of 160 trainees (Nepal, $N = 42$, Peru, $N = 82$, Uganda, $N = 36$), we conducted complete case analysis and therefore excluded those with missing data on ENACT post-tests (Peru, $N = 1$, 1%; Uganda, $N = 9$, 25%). This produced a total sample size of 150 (Nepal, $N = 42$, Peru, $N = 81$, Uganda, $N = 27$). We use paired t-tests to compare pre- to post-change of two measures of competency outcomes: the ENACT-item attribute measure (change in total harmful behavioral attributes (Level 1) and change in total helpful behavioral attributes (Level 2, 3 and 4) and the ENACT-item level measure: change from harmful level (Level 1) to nonharmful level (Level 2,3,4) and change in "competent" level (Level 3,4) from pre- to post-training. The number of attributes was

summed across all 15 competency items per participant into a total of "harmful attributes" score and a total of "helpful attributes" score and ran comparisons within and across sites. For level comparisons, we did a total count of Level 1 scores and a total count of Level 3 and Level 4 scores per trainee at pre- and post-training, within and across sites. Data is analyzed as a total sample and broken down by the three country sites. A statistical significance level of $p < .05$ was used.

### Qualitative analysis

We used framework analysis to allow for a combination of inductive and deductive modes of coding and analysis (Smith and Firth, 2011). All qualitative data were coded in Dedoose, a cross-platform application for analyzing qualitative and mixed-methods research (Dedoose, 2018). To develop the codebook, we created a theme matrix informed by the interview guide. The final codebook resulted in 10 parent codes and 5 child codes. A selection of 22 excerpts across 5 transcripts was used to check for inter-coder reliability (ICR) among 3 coders using the "Test" function in the Training Center on the Dedoose platform. After coder agreement was reached (ICR, 85%), transcripts were independently coded at the paragraph level. We used a code summary template to summarize codes and prepare for analysis. We further identified linkages and patterns in the data running queries and matrices in Dedoose. To ensure consensual validation, the coders had multiple discussions throughout the process.

### Ethics

*The authors assert that all procedures contributing to this work comply with the ethical standards of the relevant national and institutional committees on human experimentation and with the Helsinki Declaration of 1975, as revised in 2008.* The study was approved locally for each site (Nepal Health Research Council, Nepal (ERB604/2020), Universidad Peruana Cayetano Heredia, Comité Institucional de Ética en Investigatción, Peru (CIEI 19021), Research Ethics Committee (MUREC), Uganda (REF 0608-2020) and internationally by the George Washington University Committee on Human Research Institutional Review Board (IRB FWA00005945) and the World Health Organization Research Ethics Review Committee (ERC.0003437). All participants provided informed written consent to participate in the study.

## Results

### Participant demographics

Trainees included pre-service students (nursing, public health, and social work university students) from Nepal, in-service workers (community health workers (CHW), nurses, and obstetricians) and pre-service students (psychology students) in Peru, and in-service workers (CHWs) from Uganda – all which were naïve to mental health and psychosocial service delivery. Table 1 displays a breakdown of trainee characteristics.

### Training and research implementation

Each training had an average of 12 trainees, with 1–2 trainers per training. EQUIP-FHS trainings, on average, were 20 h long, typically delivered over 4 days. Supplementary Table S2 displays implementation characteristics of an average EQUIP-FHS

**Table 1.** Characteristics of trainees in EQUIP-FHS "in-service" and "pre-service" trainings (N = 13) in Nepal, Peru and Uganda

| Trainee characteristic | Nepal (n = 42) | Peru (n = 81) | Uganda (n = 27) | Total (All sites) (n = 150) |
|---|---|---|---|---|
| Age: mean (range) | 21.5 yrs. (18–30 yrs.) | 36 yrs. (20–70 yrs.) | 39 yrs. (23–73 yrs.) | 33.5 (18–73 yrs.) |
| Female N (%) | 33 (79%) | 74 (91%) | 20 (74%) | 128 (85%) |
| Occupation N (%) | | | | |
| *In-service*: | | | | 97 (65%) |
| Community health workers (CHW) | | 44 (55%) | 27 (100%) | |
| Nurses | | 15 (18%) | | |
| Obstetricians | | 11 (13.5%) | | |
| *Pre-service*: | | | | 53 (35%) |
| Public health students | 16 (38%) | | | |
| Social work students | 13 (31%) | | | |
| Nursing students | 13 (31%) | | | |
| Psychology students | | 11 (13.5%) | | |

training and number of trainings conducted in Nepal, Peru and Uganda. Each site had at least one clinical supervisor available for monitoring.

### Potentially harmful and helpful attributes

Changes in potentially harmful attribute scores and helpful attribute scores from pre-to post-training are shown in Table 2. There was a significant decrease in harmful behaviors across all sites ($t = −14.76, n = 150, p < .0001, d = −1.21$). Similarly, harmful attribute scores decreased significantly from pre- to post-training within each site: Nepal: $t = −9.65, n = 42$ $p < .0001, d = −1.49$; Peru: $t = −13.23, n = 81, p < .0001, d = −1.47$; Uganda, $t = −3.32, n = 27, p < .01, d = −.64$. There were also significant increases in helpful behaviors across all sites ($t = 16.14, n = 150, p < .0001, d = 1.32$), and within each site: Nepal: $t = 12.44, n = 42, p < .0001, d = 1.92$; Peru: $t = 13.93, n = 81, p < .0001, d = 1.55$; Uganda, $t = 3.27, n = 27, p < .01, d = .63$ from pre- to post-training. Qualitative results in Box 1 further elaborate potential reasons for such effective outcomes, such as the benefits of using a competency-based approach with practice and role plays and the relevance and importance of training in foundational helping competencies for all health workers.

### Post hoc analysis of harmful and helpful attributes

Post hoc analysis was conducted to assess whether there were any observable differences in outcomes based on the experience level of the trainees (e.g., in-service or pre-service), specifically with the Peru site dataset, as this was the only site that included both in-service and pre-service trainees. Paired *t*-tests were used to compare the pre-post change in harmful and helpful attributes by experience level, and a simple linear regression was used to test if the experience level of the trainees explained post-test scores adjusting for baseline (pre-test) scores. We found no significant difference between experience levels (e.g., pre-service being less experienced, in-service being more experienced) and outcomes. The regression results were not significant ($R^2 = .02, F(2, 78) = 1.98, p = .15$). A table showing post hoc paired t-tests can be found in the Supplementary Material.

### Harmful and "competent" levels

Table 3 shows the change in total counts of only harmful levels (Level 1) and the change in total counts of competency in Levels 3 and 4. There was significant movement from a Level 1 ("potentially harmful") score to a Level 2 ("not done or limited basic competency"),

**Table 2.** Comparison of total scores for the ENACT harmful and helpful attributes pre- and post-training

| Site | ENACT attributes | Mean (SD) Pre | Mean (SD) Post | Mean difference (95% CI) | t-statistic (df) | p-value[a] |
|---|---|---|---|---|---|---|
| All sites (n = 150) | Harmful | 3.27 (1.87) | .95 (1.47) | −2.02 (−2.52, −1.92) | −14.76 (149) | <.0001 |
| | Helpful | 9.03 (4.46) | 15.34 (5.68) | 6 (5.22, 6.78) | 16.14 (149) | <.0001 |
| Nepal (n = 42) | Harmful | 4.33 (2.19) | 1 (1.86) | −3.33 (−4.03, −2.64) | −9.65 (41) | <.0001 |
| | Helpful | 11.17 (5.05) | 20.43 (5.39) | 9.26 (7.76–10.77) | 12.44 (41) | <.0001 |
| Peru (n = 81) | Harmful | 2.83 (1.21) | .67 (1.00) | −2.16 (−2.49, −1.84) | −13.23 (80) | <.0001 |
| | Helpful | 7.98 (2.91) | 13.75 (4.00) | 5.78 (4.95,6.60) | 13.93 (80) | <.0001 |
| Uganda (n = 27) | Harmful | 2.96 (2.33) | 1.70 (1.75) | −1.26 (−2.04, −.48) | −3.32 (26) | =.003 |
| | Helpful | 8.89 (6.08) | 12.19 (5.43) | 3.29 (1.23, 5.37) | 3.27 (26) | =.003 |

[a]Paired *t*-test.

**Table 3.** Comparison of total scores for the ENACT Level 1 (Harmful) and Levels 3 and 4 ("Competent") pre- and post-training

| Site | ENACT level | Mean (SD) | | Mean difference (95% CI) | t-statistic (df) | p-value[a] |
|---|---|---|---|---|---|---|
| | | Pre | Post | | | |
| All sites (n = 150) | Level 1 | 3.31 (1.85) | .94 (1.45) | −2.37 (−2.68, −2.05) | −14.78 (149) | <.0001 |
| | Levels 3 and 4 | 0.61 (.74) | .96 (1.12) | .35 (.15, .54) | 3.56 (149) | <.001 |
| Nepal (n = 42) | Level 1 | 4.55 (2.19) | 1.00 (1.86) | −3.55 (−4.27, −2.82) | −9.87 (41) | <.0001 |
| | Levels 3 and 4 | .38 (.54) | 1.33 (1.48) | .95 (.51, 1.39) | 4.37 (41) | <.0001 |
| Peru (n = 81) | Level 1 | 2.84 (1.15) | .67 (1.00) | −2.17 (−2.49, −1.86) | −13.68 (80) | <.0001 |
| | Levels 3 and 4 | .74 (.75) | .75 (.90) | .01 (−.20, .23) | .11 (80) | .91 |
| Uganda (n = 27) | Level 1 | 2.78 (2.14) | 1.67 (1.64) | −1.11 (−1.82, −.41) | −3.24 (26) | .003 |
| | Levels 3 and 4 | .59 (.89) | 1.00 (.96) | .41 (−.01, .82) | 2.02 (26) | .05 |

[a]Paired *t*-test.

3 ("basic competency"), or 4 ("advanced competency") across all sites pre-to post-training ($t = -14.78$, $n = 150$, $p < .0001$, $d = -1.17$). Similarly, significant movement from Level 1 scores to Level 2, 3 or 4 scores from pre-to post training was found within each site: Nepal: $t = -9.87$, $n = 42$ $p < .0001$, $d = -1.62$; Peru: $t = -13.68$, $n = 81$, $p < .0001$, $d = -1.89$; Uganda, $t = -3.24$, $n = 27$, $p < .01$, $d = -.52$). There was a significant movement from pre to post-training in Levels 3, 4 (basic competency, advanced competency) from pre- to-post training across sites ($t = 3.56$, $n = 150$, $p < .001$, $d = .47$). In Nepal, participants' scores showed a significant movement with a large effect size ($t = 4.37$, $n = 42$, $p < .0001$, $d = 1.76$); however, in Peru, only a very slight movement of competency scores was made from pre-training (M = .74, SD = .75) to post-training (M = .75, SD = .90) and was not significant Peru ($t = .11$, $n = 81$, *ns*, $d = .01$). Similarly, Uganda showed a slight increase in competency scores from pre-training (M = .59, SD = .89) to post-training (M = 1.00, SD = .96), but was

insignificant ($t = 2.02$, $n = 27$, ns, $d = .46$). The average number of Level 1, 2, 3 and 4 scores across participants (N = 150) pre-and post-training can be found in Supplementary Figure S2.

Figure 2 shows the percent of participants (N = 150) that scored a Level 1 (harmful) per ENACT item at pre-and post-training. At pre-training, most participants (N = 141, 94%) scored a Level 1 (harmful) competency on assessment of harm (Item 07), of which dropped by 68% at post-training (N = 45 participants scored Level 1, 30%) at post-training. Two other areas where many participants scored Level 1 at pre-training included confidentiality (Item 03; N = 92 participants, 61%) and eliciting feedback (Item 15; N = 77 participants, 51%). These scores decreased by 82% (N = 17 participants scored Level 1, 11%) and 67% (N = 24 participants scored Level 1, 16%) at post-training. Qualitative results in Box 1 help to further explain these results wherein trainees describe more difficult competencies to achieve, regardless of their previous

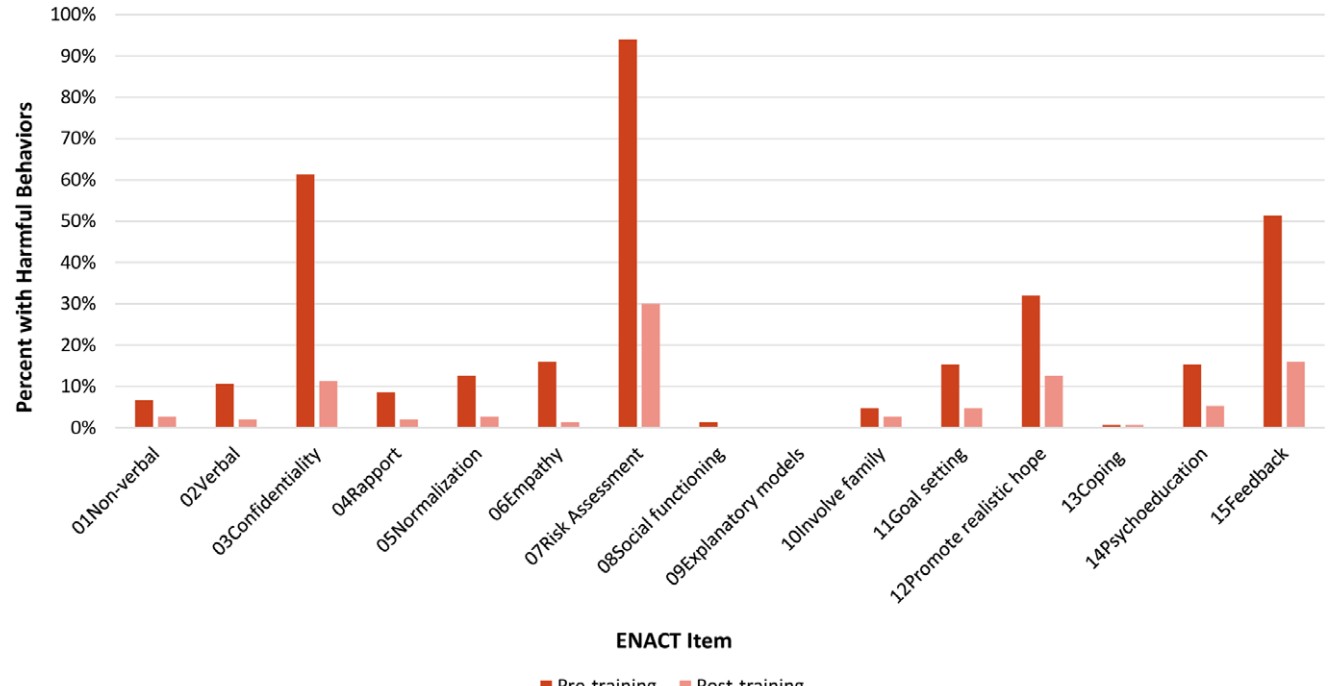

**Figure 2.** Percent of participants (*N* = 150) with Level 1(harmful) score per ENACT item, pre-and post-training.

professional training, and those which they now feel more confident in applying in their work – particularly in relation to the assessment of harm and suicidal behaviors.

Movement to competent (Level 3,4) scoring was not as prominent across all ENACT items; however, change was substantial in the items where movement was made. For instance, participants that scored a Level 3 or 4 (competent) in nonverbal (Item 01)

increased from 11 participants to 27 participants (157%) from pre-to post-training, respectively. Only one participant scored a Level 3 or 4 on assessment of harm (Item 07) at pre-training; however, this changed to 6% ($N = 9$) of participants scoring as competent in assessment of harm (Item 07) at post-training. Qualitative results in Box 1 elucidate these findings, wherein trainees describe competencies that were most difficult to learn or achieve.

---

**Box 1.** Key highlights from qualitative interviews with trainees and trainers on FHS training

Across sites, assessing suicidal behavior (ENACT07), goal setting (ENACT11), empathy (ENACT06), and responding to feelings and normalization (ENACT05) were mentioned most often as competencies that took more time to learn:

*The biggest challenge was the topic of suicide…As little knowledge as I may have had, I knew what questions could be asked based on what I had read and what we had been trained in "Noguchi" [National Mental Health Institute Honorio Delgado-Hideyo Noguchi] about suicide. So, it never occurred to me to talk about it directly with the patient…Even during role-playing it was a little bit difficult to ask those questions, which were quite sensitive; I didn't know that getting straight to the point helped [patients] the most. – Obstetrician 1, Peru*

*Most challenging…setting the goal, yeah. Like [using] the actual words, you are not supposed to force someone. – CHW 2, Uganda*

*More challenging was like … suicidal thoughts… to deal with such patients … and we were so conscious like, will we be able to use the appropriate word during that counseling period or not? I think when that type of situation arrives, then that is the challenging one. – Nursing Student 1, Nepal*

Some trainees described how they had applied their skills since finishing the training, including the changes they saw in their clients:

*The training taught me to have a lot more patience and to emphasize a lot more issues. For example, I didn't have the habit of bringing up the topic of suicide. Now, this week [at work], I'm doing it and I'm realizing that a lot of patients don't tell you right away. But [then]…at night, or after the session, they are sending me messages asking if they can talk to me and "it's something personal." So, it's helped me a lot. – Obstetrician 2, Peru*

*…in the hospital setting, it [the foundational helping competencies] is very helpful. During the case presentation, we take the history of patients… And while taking the history, we have to maintain the IPR [interpersonal relationship] and for that, we have to listen to them properly and talk to them in a good way. – Nursing Student 1, Nepal*

*…I feel like more patients are going to come to me, especially being referred by fellow patients, because once you handle someone better, they… recommend you to another person, not because you have given them a drug and they healed, but because of the way you handled them in the process…Also [I] am going to be fulfilled because I know I did the right thing, not just because of the profession, but as a human being. – CHW 4, Uganda*

Trainers compared differences, and perceived benefits, of the EQUIP-FHS competency-based training to other trainings:

*In other trainings, mostly we took it alongside the theory, from the background, everything [theoretical] would be included, whereas in this [training], there was more of an introduction given on the contents briefly, and then practice was given more focus… – Trainer 2, Nepal*

*The first few days are somehow shocking to them [trainees], because they were not used to the assessment, or the role plays, or showing them the results, or putting the skills into practice, because most of their trainings had been theoretical with a theoretical assessment… The day after the assessment they received feedback, and they sort of adapted to this methodology. [Then] they felt more prepared, more confident, compared to the first few days. – Trainer 2, Peru*

*As a clinical psychologist, if I had continued the way I was without FHS [competency-based training] I wouldn't be able to help, because there are so many things that I would not bother about. The sweetness is the practical attributes and pointing out how those attributes have an interplay in the person that you're helping [training], because it demonstrates the good and the bad behaviors…this has created awareness now that each time a bad behavior is trying to cross your way, you're conscious. – Trainer 1, Uganda*

Trainers and trainees described challenges to training, both remotely and in-person:

*When it's an online format, sometimes there is problem because of poor internet connection and sometimes the electricity gets cut off. Because of that sometimes we might miss the important things from the training. – Social Work Student 1, Nepal*

*My biggest challenge has been dealing with time, motivation…I am talking specifically about the nurses and the obstetricians…most of them were working, especially the nurses were constantly working, they didn't have 100% of their time available to receive the training… if necessary, [I know we can] add more hours, more days to the training…But the issue was the willingness of the participants. – Trainer 2, Peru*

Trainees recommended organizations or professions that could benefit from an FHS training:

*On the first day of the assessment, I mentioned that I would like all my colleagues to take this course. These skills that we have all learned on how to treat and care for our patients should be shared with others so patients can be cared for in an adequate manner. Not just one group, but all of us, so that everyone goes through this type of training. – Nurse 2, Peru*

*I recommend it [EQUIP-FHS] because many people like…local leaders, the Local Councils, the politicians, they are missing the skills. And even the political leaders, the way they are handling their offices, they have to get that knowledge of helping, the foundational helping [competencies]…to help the people who come to their offices. – CHW 6, Uganda*

Trainers described pathways and potential blockages to scaling up EQUIP-FHS training:

*I believe one of the barriers to scaling up this training, at least in the public sector, is the issue of working hours and productivity levels…For example, in the case of obstetricians, the person in charge of the sexual health unit told me: "If she participates in your training, she will stop taking care of pregnant women. That lowers my productivity, lowers my indicators." I told her she was right…not having staff available for [4 daily hours for 1 week] can affect users and patients, but at the same, it would generate a benefit. – Trainer 2, Peru*

*If one involved leadership, then the demand will get higher than the supplies… we need to have trained trainers, so we'd have to have a ToT [training of trainers] …We must do a sensitization of the leadership…you know, give a presentation to them and justification why we think it's necessary. And then ask them to nominate people to be trained…maybe heads of department… – Trainer 1, Uganda*

A visual showing the percent of participants (*N* = 150) that scored a Level 3,4 (competent) per ENACT item at pre- and post-training can be found in the Supplementary Material.

### Qualitative interviews

A sample of trainees and trainers from Nepal, Peru, and Uganda were interviewed between December 2020 and November 2021. They reported their experiences and perceptions with participating in the FHS trainings within 1 week of finishing the training. Additionally, some trainees reported how they were applying what they had learned from the FHS training into their daily work 3–4 weeks post-training. Some extracts from their accounts can be found in Box 1.

### Discussion

To reach the United Nations (UN) Sustainable Development Goal (SDG) 3.8 (*achieving universal health coverage, UHC*) targets by 2030, organizations around the world agree that health systems need more competent and empowered workers (World Health Organization, 2019a, 2019b, 2022; Rees et al., 2021; Ssengooba et al., 2021; Adhikari et al., 2022). Locally and globally, existing competency frameworks identify foundational helping skills as core to the minimum standards healthcare workers and other service providers need to ensure safe and effective care. However, as the results from this study show, these training programs may not currently be achieving their competency goals: at pre-training, 98% of healthcare workers and other service providers (*N* = 148) scored Level 1 (harmful) competency on at least one ENACT item, while on average, these trainees scored as potentially harmful on 3 of the 15 ENACT competencies. It is possible that the education these currently practicing healthcare workers and other service providers received before the EQUIP-FHS training either did not cover these competency topics, or if the education did cover these topics, it may have been strictly knowledge-based learning. This latter connection was identified in the qualitative results, wherein in-service workers compared their previous professional training to EQUIP-FHS training in building competency (e.g., knowledge, attitudes and skills). For instance, some pointed out that the techniques used in the EQUIP-FHS training increased their competency in the assessment of risk of harm, such as suicide risk, as compared to what was taught in the existing National Mental Health Institute Honorio Delgado-Hideyo Noguchi training in Peru that gave them "little knowledge."

Similarly, competency may be inadequately assessed in existing healthcare worker and other service provider education programs when using unidimensional measures, like multiple-choice knowledge tests, that cannot capture complex, multidimensional constructs such as competency (DeVellis, 2012). On a programmatic level, essential information to inform and improve educational curricula or training programming may be missed when using norm-referenced results, such as from knowledge tests (Tucker, 2015), and could increase the inaccuracy of results due to the biases in formatting, language, and generalization of indicators to meet a "norm" population (Underly, 2021), thereby increasing ambiguity for service delivery monitoring and evaluation purposes. Experts have suggested a more multidimensional approach to measuring competency through observational roleplay assessment (Miller, 1990; Tucker, 2015; Mills et al., 2020), which could be why this study captured such detail on harmful behaviors.

This study provides evidence that competency-based training in foundational helping can be implemented in a brief time (~20 h). It also showed that the EQUIP-FHS competency-based tools for assessment and providing feedback were effective in building and assessing competency from pre- to post-training and found to be feasible, acceptable, and useful among trainees and trainers from Nepal, Peru and Uganda. We contacted the EQUIP-FHS training participants up to 3 weeks post-training to qualitatively determine how they were applying the foundational helping competencies in their daily work. Specific examples were provided by nurses, obstetricians, community health workers and students on how these competencies were having a positive impact on their work. Long-term follow-up competency assessment (e.g., 3, 6, 12 mos.) is needed to understand the extent to which these competencies are retained, as well as how much and how often an EQUIP-FHS competency-based training may be needed (e.g., refresher trainings). Similarly, impacts on healthcare and other service delivery should be investigated, as one community health worker in Uganda thoughtfully requested post-training, "*I would love to see my progress (from this training). Looking at the number of cases, the number of clients that I have handled and checking on to see if they are satisfied, if they are happy, the guidance I gave them, the help I gave them…*"

The EQUIP-FHS curriculum is modular and each site had a similar selection of modules in this study. Six ENACT competency items (social functioning (Item 08), explanatory models (Item 09), family involvement (Item 10) and coping mechanisms (Item 13), psychoeducation (Item 14) and eliciting feedback (Item 15)) were not specifically included in the training plan but deemed optional and therefore covered in less detail by the trainers. This implementation approach likely explains why only a few trainees' competency scores change from pre-post training in these six items. Future organizations and governmental bodies implementing EQUIP-FHS training are encouraged to use a similar modular approach, mainly by optimizing competency assessment results to determine which modules to deliver. Based on results and trainer feedback, we recommend that two trainers support the implementation of any EQUIP-FHS training plan to optimize the use of observational assessments and competency-based feedback.

The findings of this study show that the current EQUIP-FHS training manual is applicable and easy to use across three different settings when translated into the language in which the training will be delivered. We strongly encourage organizations to access the translated and adapted mini roleplay videos, roleplay scripts, and assessment tools, freely available on the EQUIP platform (https://equipcompetency.org), for direct implementation or for replicating these materials into a different language. Further guidance for implementing the EQUIP-FHS training package is being developed, including how trainers can contextually adapt role plays and other activities by actively involving the trainees, like asking about their daily work activities and role playing these scenarios when practicing foundational helping competencies. This approach could strengthen learning and help trainees apply the competencies to their current or future work.

Given the brevity of the training along with perspectives highlighted in our qualitative data, it is reasonable to expect that more EQUIP-FHS training time (>20 h) could increase the number of healthcare workers or other service providers who score competently (Level 3,4) post-training. Likewise, pre-service and in-service workers with previous experience in mental health and psychosocial services may be expected to reach higher competency levels within a shorter time frame (~20 h). As such, organizations and

governmental bodies implementing EQUIP-FHS or similar EQUIP competency-based approaches should consider the advantages and disadvantages of increasing the time spent training workers with various occupational backgrounds.

## Recommendations

As part of progress toward achieving universal health coverage (SDG 3.8), we recommend implementing and scaling the EQUIP-FHS training for in-service and pre-service workers across healthcare and other service sectors. Based on the findings of this research, we present various implementation strategies.

1. *Trainers assess competencies to strive for a "do no harm" approach among healthcare workers and other service providers*

To truly verify that client safety is at the forefront of training and supervision program outcomes, focus needs to shift from using only performance measure indicators at the organizational level (e.g., incident reporting, productivity) to using behavioral assessments at the individual level (e.g., trainers assessing healthcare worker and other service provider behaviors). Results from this study suggest that competency-based training like EQUIP-FHS offers the tools organizations and governmental bodies need to ensure their programs are achieving the "do no harm" principle. Notably, results from this study show that pre-service and in-service workers naïve to mental health or psychosocial service delivery (e.g., nurses, obstetricians) should ideally demonstrate no harmful behaviors and aspire to competency in foundational helping. To integrate competency-based assessment and approaches into programs, investment is needed in training trainers and supervisors on how to use competency-based techniques like rating competency assessment tools, incorporating role plays, providing competency-based feedback, and adjusting training and supervision plans accordingly. In this way, trainers and supervisors can actively identify harmful behaviors (Level 1) during training and supervision sessions and provide tailored feedback or additional training to help trainees remove these behaviors and replace them with more helpful behaviors.

2. *Integrate competency-based training and assessments into pre-service programming to strengthen the quality of a growing workforce across healthcare and other service sectors.*

The qualitative findings in this study suggest that it could be challenging to find time for in-service workers to participate in an EQUIP-FHS 20-h curriculum. As such, we recommend this training during pre-service as an optimal pathway for scaling competency-based approaches and to build foundational helping knowledge, attitudes, and skills. An evidence-based competency assessment complements each module in EQUIP-FHS, and the digital tools on the EQUIP platform provide immediate results to help trainers and supervisors provide tailored feedback. Such techniques are ideal for implementing into pre-service programs and curricula. For instance, EQUIP-FHS competency-based training could be part of an initial orientation for first-year medical residents. Thereafter, competency assessment, feedback, and refreshers could continue throughout the program.

Similarly, competency-based assessments and feedback could be paired with existing program modules or curricula and are optimally used during pre-service supervision sessions. In Uganda, partners from Makerere University suggested integrating EQUIP-FHS modules into the community, counseling and clinical psychology training programs for undergraduate and graduate students as part of their pre-service curriculum (Alipanga and Kohrt, 2022). Additionally, they plan to integrate competency assessments and feedback with their students throughout their programs and train second-year undergraduate and graduate students as competency raters to support a feasible scale-up of this approach. In Nepal, partners at TPO Nepal have begun using the EQUIP-FHS training with nonspecialists as pre-requisite course to training on manualized interventions, such as the WHO's Thinking Healthy Program (THP) (see Supplementary Material for sample schedule) and Problem Management Plus (PM+).

Furthermore, as the identity of the healthcare and service provider workforce expands to include a range of cadres that directly engage with people and offer a helping role, such as seen through the increase in the training of police officers to appropriately and safely respond to people dealing with mental health problems, or the training of barbers to be mental health advocates, an EQUIP-FHS training could ideally be integrated into police academy training or barbershop school to support these and similar efforts.

3. *Work with leaders, working groups, and public health structures to integrate competency-based assessment and feedback for strengthening in-service care delivery, monitoring and evaluation, including task-shared programs.*

According to feedback from trainers and trainees in this study, EQUIP-FHS competency-based training would benefit "all walks of life," including workers and the persons they engage with across health and other service sectors (e.g., doctors, teachers, police officers, political leaders). Additionally, trainers and trainees promoted the scale-up of EQUIP-FHS competency-based training and assessment and suggested leveraging leadership and managing perceived barriers in public health structures to do so. The competency-based approach to EQUIP-FHS supports its modularity, and as such, there is flexibility to which elements of EQUIP-FHS and its competency-based techniques are incorporated. Partners at Socios En Salud in Peru have been generating alliances with various institutions in the public and private health sector, including the Ministry of Health, to support continued implementation of EQUIP-FHS training and general use of EQUIP competency-based assessment and feedback in programs for various public health professionals. For instance, competency-based assessment can be included as part of ongoing health facility and service utilization monitoring and evaluation, whereby a program manager, facility manager or supervisor incorporates competency assessments during observation of real-world care delivery and client engagement. Such incorporation could provide invaluable, detailed feedback on the performance of healthcare workers and other service providers and the evaluation of existing training protocols.

Another tactic to help healthcare workers and other service providers retain their foundational helping competencies could be to hold single module-based refresher trainings as an approach to continuing education. Similarly, "practical" homework assignments could be integrated into continuing education. Trainers in Peru successfully assigned "practical" homework to obstetricians and nurses who were participating in EQUIP-FHS training during the evening while still working during the daytime to deliver care. The trainers asked the obstetricians and nurses to practice or apply the daily learned foundational helping competencies at work and then discussed how that application went during EQUIP-FHS training hours. In Uganda, HealthRight has adopted the EQUIP competency-based training approach and competency tools within their programming, including assessing staff delivering WHO's

PM+. They are also promoting competency assessments and competency-based training throughout their mental health and psychosocial support (MHPSS) national working groups and will conduct training for the Uganda Counsellors Association, including professional counselors working in various practice settings.

## Limitations

This study cannot account for differences in trainer styles or training modes (e.g., in-person, virtual), which may contribute to competency outcomes. The uncontrolled before-and-after design lacks the robustness of a randomized experimental design and has limitations, including certain threats to validity because of several biases (e.g., regression-to-the-mean, history, instrumentation, testing, maturation, dropout, placebo, Hawthorn). We believe these biases were minimized as follows: regression-to-mean bias was low-risk because we did not restrict our recruitment of trainees based on any characteristics, but allowed any type of in-service or pre-service participant take part in the training as long as they had no experience delivering mental health or psychosocial services. Our study unlikely experienced maturation or history effects due to the short time frame in which EQUIP-FHS trainings were implemented. We had high follow-up rates (all sites: 150/160, 94%; Nepal: 42/42, 100%, Peru: 1/82, 1%; Uganda, 9/36, 25%), except for the Uganda site, of which the dropout rate was still relatively low. Regarding practice effects (test–retest), we do not anticipate that the repeated ENACT role plays (pre- and post-training) outside the training context introduced a change in competency because formal feedback from these assessments was not provided, and participants were given different role plays scenarios at each time point. Placebo and Hawthorne threats to bias were very low, as this study was not clinical in nature, and the researchers collecting the data within each site were not involved in any way with the trainings. Finally, instrumentation was consistent for pre-post data collection.

Although inter-rater reliability (IRR) exercises were conducted in each site, different strategies to measure ICCs varied depending on whether groups of raters rated the same participant. An advantage to this study was calculating a summary score across raters per site, which may be a more realistic approach to ensure the reliability of ratings with a nonparametric observational tool in real-world settings compared to controlled research environments. Finally, the follow-up period was very short across sites (1 day to 3 weeks). Longer follow-up periods on how trainees continue to apply foundational helping competencies in their daily work are needed to understand how much and how often training (e.g., refresher trainings) is needed.

## Conclusion

A brief competency-based curriculum in foundational helping knowledge, attitudes and skills, delivered through pre-service or in-service training, can reduce the risk that healthcare workers and other service providers display harmful behaviors. The EQUIP-FHS training curriculum provides evidence that competency-based assessment, feedback and other competency-based approaches can help ensure that in-service and pre-service workers are meeting their competency goals and ultimately engaging with clients safely. Trainers, supervisors, program managers, leaders in non-profit, governmental, and academic organizations, and donors are

encouraged to use this study's training materials and results to guide future program planning, evaluation and funding.

**Open peer review.** To view the open peer review materials for this article, please visit http://doi.org/10.1017/gmh.2023.43.

**Supplementary materials.** The supplementary material for this article can be found at http://doi.org/10.1017/gmh.2023.43.

**Data availability statement.** The data that support the findings of this study are available from the corresponding author, G.A.P., upon reasonable request.

**Acknowledgments.** We are grateful to the team members of the EQUIP partner teams who helped: Nepal (FHS trainer: Indira Pradhan; Collected data: Bimala Shrestha, Jagat Mohan, Sunil Khanal, Kumar Poudel, Ujwal Bhandari, Niko Gautam); Peru (FHS trainer: Margot Aguilar, Janeth Santa Cruz; Collected data: Maricielo Espinoza, Margaret Rivas) and Uganda (Collected data: Samuel Wasekera, Joel Omoding, Ayio Rebecca, Emmanuel Oboi, Sharon Charity Akello). From GWU, we also thank Jenny Sabol and Lea Simms for their help on qualitative coding, and Allison Morrow for her support in cleaning quantitative data.

**Author contribution.** Conceptualization: G.A.P., A.S.; B.A.K.; Data curation: G.A.P., A.S., L.M., D.R.; Formal analysis: G.A.P.; Funding acquisition: A.S., B.A.K.; Investigation: P.S., J.A., A.S., L.M., V.N.; Methodology: G.A.P., P.S., J.A., A.S., B.A.K.; Project administration: G.A.P., P.S., J.A., A.S., B.A.K.; Resources: G.A.P., B.A.K.; Supervision: N.P.L., R.K., C.C., J.T.G., A.S., B.A.K.; Visualization: G.A.P.; Writing – original draft: G.A.P.; Writing – review and editing: P.S., J.A., A.S., N.P.L., R.K., C.C., J.T.G., L.M., V.N., D.R., A.S., B.A.K.

**Financial support.** This study was possible due to the Ensuring Quality in Psychological Support (EQUIP-1 and EQUIP-2) project, which was funded by the World Health Organization and the United Nations International Children's Emergency Fund (UNICEF) through a grant from USAID. B.A.K. is supported by the U.S. National Institute of Mental Health (R01MH120649).

**Competing interest.** A.S. is employed with the World Health Organization. All other authors declare no competing interests.

**Ethics standard.** All authors declare to adhere to the publishing ethics of Global Mental Health.

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
