## [Reviewer Report]

Dear Drs Judy Bass and Dixon Chabanda, Editors-in-Chief:

We wish to submit an original research article entitled “A mixed methods evaluation of a World Health Organization competency-based training package for Foundational Helping Skills among pre-service and in-service health providers in Nepal, Peru and Uganda” for consideration by Cambridge Prisms: Global Mental Health. We confirm that this work is original and has not been published elsewhere, nor is it currently under consideration for publication elsewhere.

In this article, we report quantitative and qualitative results on the evaluation of 13 trainings using a competency-based curriculum to teach 150 pre-service and in-service health providers foundational helping skills (FHS) in Nepal, Peru and Uganda. The results show significant impact in reducing harmful behaviors and increasing helpful behaviors, as assessed by the ENhancing Assessment of Common Therapeutic factors (ENACT) tool on the WHO/UNICEF Ensuring Quality in Psychological Support (EQUIP) digital platform (www.equipcompetency.org). 

Programs globally are aiming to reach Universal Health Coverage in 2030, of which requires quality training of health workers to ensure they have the core skills needed to deliver supportive and safe care. Using a competency-based approach to training and supervision in FHS that is in line with global and local competency frameworks has the potential to support academic institutions, governments and organizations in reaching these goals; however, curriculums are lacking. This study is significant because it reveals the effectiveness of a WHO brief competency-based training package to reduce harmful behaviors and increase helpful behaviors related to FHS among both in-service and pre-service providers. Also, this study highlights the feasibility and acceptability participating and delivering the training, and the benefits of scaling-up competency-based programs as described by providers and trainers from three lower-resourced settings. The WHO/UNICEF EQUIP platform was launched in March 2022 to offer freely available competency assessment tools, e-Learning materials, and other resources to anyone around the world. The evaluation in this study is contributing to the final publication and global release of the manualized WHO EQUIP competency-based curriculum for trainers and supervisors to teach FHS, predicted to be released by Fall 2023. 

We believe that this manuscript is appropriate for publication by Cambridge Prisms: Global Mental Health because it offers insight into potential pathways for implementing and scaling competency-based programs that offer the tools needed to assess health workers in various sectors (e.g. psychiatry, psychology, nursing, obstetrics, public health) are doing no harm. Based on our results, we highlight the application and usefulness of incorporating the EQUIP FHS competency-based curriculum into academic institutions for pre-service training of various social and health workers for a feasible approach to strengthening the future in-service workforce. We offer key recommendations for institutions for implementing competency-based approaches like the FHS curriculum into their existing programming. 

We anticipate this could be a highly cited manuscript as application of the tools and resources on the EQUIP platform, and training in FHS, continues to build globally. 

One author of this article is currently employed with the WHO; all other authors have no conflicts of interest to disclose. All authors confirm that they fulfill the criteria listed for authorship by ICJME recommendations, and that no authors who would reasonably be considered an author have been excluded. 

For reviewers, we recommend Gilles Dussault, Freddie Ssengooba, Gareth H. Rees, and Zulfiqar A. Bhutta. This evaluation was possible with funding by the World Health Organization and the United Nations International Children’s Emergency Fund (UNICEF) through a grant from USAID. The senior author, Brandon Kohrt, is supported by the U.S. National Institute of Mental Health (R01MH120649). 

Gloria A. Pedersen, DrPH, MSc

Senior Research Associate, The Center for Global Mental Health Equity

George Washington University School of Medicine and Health Sciences

2120 L St NW Suite 600, Washington DC 20037 

gapedersen@gwu.edu

---

## [Reviewer Report]

GMH-23-0092

This paper describes the outcomes of a foundational helping skills training for paraprofessionals. I appreciate the inclusion of qualitative and quantitative data as well as the fact that this paper is framed around universal health coverage – these skills are important for all providers, not just those working in mental health. Additionally, the fact that data are collected across three different contexts allows this paper to highlight commonalities and differences across sites. I have made some recommendations below to further refine this paper, largely focusing on the integration of quantitative and qualitative methods.

- Abstract

o I find the use of the word educated to be a little confusing in that first sentence (this is also used in the impact statement). It leads me to expect a study focused on increasing knowledge, not skills. Maybe replacing that with skills or competence would be more appropriate?

- Introduction

o It may be helpful to provide a little more context on why it’s important to increase FHS skills among providers. What are the consequences to patients and health systems if providers do not have these skills?

o I may have missed it, but I don’t believe MHPSS is defined anywhere.

- Methods

o Was the e-learning module on giving and receiving feedback for supervisors required or optional?

o Please provide a brief explanation as to why the sample from Peru was so much larger.

o I appreciate that you focused on a clinically meaningful outcome – the change from harmful to non-harmful.

□ Did you run statistical tests for the change in levels for individual items or just calculate proportions harmful and not harmful pre and post?

o Can you speak to the cultural adaptation and/or translation decisions made in each of these three contexts? For example, why were those specific items excluded or made optional (as mentioned in the discussion)?

- Results

o Because this is a mixed methods paper, I was expecting an integration section to explicitly relate the quantitative and qualitative findings to other another. And I do think such a section would be helpful. For example, there seem to be parallel findings on the risk item across quant and qual.

□ This reference may be helpful in deciding upon a method of integrating results: https://www.bmj.com/content/341/bmj.c4587.long

o Did you observe any differences across sites based on the experience level of trainees?

- Discussion

o It was not clear to me what you were saying about the optional modules aligning with competency results (line 349). Could you rephrase?

o Again, could you discuss the need to culturally adapt this curricula based on the setting?

o I think it would be helpful to link the recommendations more clearly to the common challenges experienced across sites in the qualitative work. An integration section in the results (as I describe above) would build nicely into that.

---

## [Reviewer Report]

This is a quasi-experimental study (pre-post design) describing the use of the WHO EQUIP platform to evaluate the competency in basic ‘helping’ (counselling) skills of non-specialist mental health providers following training in 3 countries. This is a well-written manuscript and gives a good demonstration of the use of the platform. It should be interest to the global mental health audience engaged in provision of psychosocial care to people with common mental disorder.

---

## [Reviewer Report]

My congratulations to the authors, and those who have developed and delivered the training package. My feedback relates to the write-up of the training package, rather than the course itself.

Overall, I found that the paper makes some bold claims that aren’t necessarily backed up by the information provided; and it would benefit from some restructuring of the text, to support the recommendations – and perhaps identify new ones, for example the logistics or considerations in operationalizing the course. The recommendations are also framed more generalizable than FHS or this course, but as relevant to the whole of health worker education – is that intentional, and can that be justified based on the study?

I find the discussion of ‘harmful’ and competence to relate to two different concepts, and this would benefit from clarification. Was the assessment tool established to assess not only ‘not competent’ but also ‘harm’? There is a frightening level of ‘harm’ reported – 98% of learners – what is the implication for other health workers who have not completed the training? But then I wondered, is it harm, or is it not yet learned competence?

I would also encourage consistency in the language throughout. I am familiar with the many different terminologies and definitions for competence, competency and plurals of the same. Defining up front what is meant by each term would help – competency levels and competent levels are used interchangeably for example; but it also seems that competency can refer to skills. Similarly, there is reference to healthcare workers, health workers and health and social care workers, as well as in many places, providers; this again would benefit from streamlining and consistency.

More specific observations are listed here:

Line 6 – add MHPSS acronym, as this later used

Line 9 – Universal instead of United.

First para – check the intent; not accurate to state that the FHS are part of the Global Competency Framework for UHC, as they are not identified as such – worth stating ‘aligned with’ or other overlapping content? It would be worth expanding this paragraph to establish the rationale and need for FHS – as it does not come directly from the UHC framework.

First para – important to situate these for mental health services, rather than for all health services – the skill ‘communication for hope’ is not consistent across all health services, but is important for mental health.

Harmful and not competent used to describe levels 1 and 2. Conversely, I find that the opposite of competence is not always incompetence. Was the assessment designed to differentiate between ‘harm’ and ‘not competent’? If not, then I would suggest reframing the narrative after Table 2 around ‘competence’

Line 331 – first mention of social services. It may be simpler to align with UN terms and refer to ‘health and care workers’ – nb WHO doesn’t use the term ‘provider’ as it is used in some country contexts to refer to the health workers, and in order countries, to refer to the organizations.

Line 332-333 – check, the implication is that the learners who were test pre-training had completed previous training programs that are deemed to be not achieving their goals. Can the evaluation of those courses be inferred from the pre-training test of individuals?

Line 336 – only competency-based assessment is linked to verifying competence. How that learning has been acquired – whether through formal training programmes or otherwise – is not necessary here.

Lines 227-343 strike me as being conclusions, not discussions. Suggest the weblink and availability is provided elsewhere. Additional detail about the performance metrics of the assessment items, as well as post-assessment transfer to practice and retained knowledge e.g. 6 and 12 months post training, must be provided. Otherwise, the claim should be tempered to reflect what is assessed – pre- and post-test score change for example. There is also a big debate about ‘measurement’ of competence, which implies precision – the authors might thoughtfully consider their use of the term.

Lines 344-358 would benefit from clarity. The first sentences are providing new context about the course, which should be established before the discussion. But it is then unclear what is meant by ‘this’ in line 348 – what is reflected in the re4sults?

The rationale supporting recommendation 1 doesn’t seem to support or relate to recommendation 1. I would expect the discussion to reflect that ‘level 1’ is worse than not providing care, and therefore what actions to take if a learner does score Level 1. There is then discussion around the practicalities of course provision, for example training trainers and supervisors which seems irrelevant to the recommendation about do no harm, or the discussion about length of time etc. This could be considered for a new/separate recommendation 4.

Line 390 – what is meant by ‘pre-service programming’. The example in line 397 of police academy does not fit the accepted definition of ‘health worker’ used in the wording of the recommendation.

It is unclear who the recommendations are for. How can the training be integrated during in-service care – does this mean supervised practice, or in-service learning? Who should integrate competency-based assessments or work with public health structures? Are these recommendations applicable beyond FHS and in fact relevant to the whole of pre-service and in-service learning programmes?

---

## [Reviewer Report]

Dear authors:

Please, check the comments of the reviewers and make the suggested corrections accordingly.

Thank you.

---

## [Reviewer Report]

Dear Dr Dixon Chibanda,

We appreciate the feedback and are glad to submit a revision to this manuscript. We have included a tracked-changes version of the manuscript, an updated supplement file, along with a response to reviewers point-by-point response. Please let us know if you will need anything more to support this resubmission.

---

## [Reviewer Report]

Dear Authors:

The reviewers have accepted your corrections to the manuscript.

The similarity anti-plagiarism software reports some problems within your text.

Please check the manuscript with your own software and correct the text accordingly.

Thank you.